# Enhanced ResNet-50 for garbage classification: Feature fusion and depth-separable convolutions

**Lingbo Li[1]☯, Runpu Wang[2]☯, Miaojie Zou[3], Fusen Guo[4], Yuheng Ren[5]\***

**1** Library of Information Center, Zhejiang Technical Institute of Economics, Hangzhou, China, **2** School of Computer Science Engineering, University of New South Wales, Canberra, Australia, **3** Faculty of Business and Economics, Monash University, Melbourne, Australia, **4** School of Systems and Computing, University of New South Wales, Canberra, Australia, **5** School of Business Economics, European Union University, Montreux, Switzerland

☯ These authors contributed equally to this work.
\* hoture@126.com

**Data Availability Statement:** All relevant data are available on Figshare:https://figshare.com/s/c817706ed7cbc8631fe5.

**Funding:** The author(s) received no specific funding for this work.

## Abstract

As people's material living standards continue to improve, the types and quantities of household garbage they generate rapidly increase. Therefore, it is urgent to develop a reasonable and effective method for garbage classification. This is important for resource recycling and environmental improvement and contributes to the sustainable development of production and the economy. However, existing deep learning-based garbage image classification models generally suffer from low classification accuracy, insufficient robustness, and slow detection speed due to the large number of model parameters. To this end, a new garbage image classification model is proposed, with the ResNet-50 network as the core architecture. Specifically, first, a redundancy-weighted feature fusion module is proposed, enabling the model to fully leverage valuable feature information, thereby improving its performance. At the same time, the module filters out redundant information from multi-scale features, reducing the number of model parameters. Second, the standard 3×3 convolutions in ResNet-50 are replaced with depth-separable convolutions, significantly improving the model's computational efficiency while preserving the feature extraction capability of the original convolutional structure. Finally, to address the issue of class imbalance, a weighting factor is added to the Focal Loss, aiming to mitigate the negative impact of class imbalance on model performance and enhance the model's robustness. Experimental results on the TrashNet dataset show that the proposed model effectively reduces the number of parameters, improves detection speed, and achieves an accuracy of 94.13%, surpassing the vast majority of existing deep learning-based waste image classification models, demonstrating its solid practical value.

**Competing interests:** The authors have declared that no competing interests exist.

## Introduction

With the rapid economic development and continuous improvement of living standards, the types and quantities of waste generated are rapidly increasing. Improper waste management, including careless disposal, incineration, and landfilling, poses a severe threat to human health and causes significant harm to the environment on which humanity depends for survival [1]. According to a 2018 World Bank report, approximately 242 million tons of plastic waste were produced globally in 2016, accounting for 12% of the world's total solid waste. By 2050, global waste production is projected to reach 3.4 billion tons annually, a significant increase from the current 2.01 billion tons [2]. Data shows that the total amount of waste generated rises yearly. If not dealt with by effective management measures, the environmental impact will be catastrophic and difficult to reverse [3].

Given the above, there is an urgent need to study a reasonable program to classify garbage effectively. The correct classification of garbage can bring multiple benefits to the plants and animals on earth as well as human life [4], which are mainly reflected in the following aspects: (1) Recycling part of the recyclable garbage can be recycled, thus reducing the amount of landfill by at least nearly 60%, thus effectively improving the efficiency of land use (2) Reducing the pollution of wastes, protecting the ecological environment, and improving the air quality. (3) Effectively recycle renewable resources. In daily life, 30% to 40% of waste can be recycled, and the recovery of these renewable wastes helps to increase the recovery rate and secondary utilization of resources. Therefore, it is crucial to reasonably classify and dispose of waste to achieve minimization, harmlessness and resourcefulness [5].

With the advancement of Internet technology and the rise of artificial intelligence, machine learning [6–9] and deep learning [10–13] have been rapidly developed and widely used in image classification. However, although both methods have achieved concrete results, each still has shortcomings. Machine learning requires manual design and feature selection, and capturing high-level semantic information from images is often challenging, thus limiting classification performance [14]. In addition, due to the small size of existing trash image datasets, machine learning and deep learning in trash image classification generally face problems such as low classification accuracy and lack of robustness. Meanwhile, the model size and detection speed need to be further optimized. Therefore, research on garbage image classification needs to explore more advanced and practical solutions.

This paper proposes a new garbage image classification model that uses ResNet-50 as the backbone network. In this model, a redundancy-weighted feature fusion module is proposed, combined with depth-separable convolution techniques to optimize ResNet-50. These two techniques effectively reduce the number of model parameters while improving classification efficiency. Additionally, the standard Focal Loss is weighted to mitigate the impact of class imbalance on model performance, enhancing the model's robustness. Experimental results on the TrashNet dataset show that the proposed model significantly improves classification accuracy and robustness while maintaining fewer parameters and faster detection speed. The main contributions of this paper are as follows:

1. This paper proposes a Redundancy-weighted feature fusion module, aiming to reduce the computational cost of the network by removing duplicate information from multi-scale features. At the same time, the weight coefficients of feature information at different scales are weighted to ensure that the model can fully use valuable feature information for the task during the feature fusion process to improve the expressive ability and classification accuracy of the model.

2. In this paper, the standard 3×3 convolution in ResNet-50 is replaced with depth-separable convolution to reduce the number of parameters and computational complexity of the model. This improvement not only preserves the feature characterization capability of the original convolutional structure but also significantly improves the computational efficiency of the network, making the model more suitable for resource-constrained devices or environments.

3. In order to cope with the problem of category imbalance, we add a weighting factor to the Focal Loss. This design not only can effectively deal with samples that are difficult to classify but also alleviates the negative impact of category imbalance on model performance while ensuring the robustness of the overall model.

## Related works

Researchers have recently proposed a series of garbage image classification methods, usually categorized into two main groups: machine learning-based models and convolutional neural network-based models.

### Garbage image classification method based on machine learning

TrashNet data set, as a publicly available garbage classification data set, has attracted many researchers to adopt various machine learning algorithms to optimize classification accuracy. Yang et al. [15] adopted support vector machine (SVM) technology and achieved 63% classification accuracy. Costa et al. [16] used random forest (RF) and extreme gradient boost (XGBoost) algorithms to achieve an accuracy of 62.61% and 70%, respectively. Then, Satvilkar et al. [17] adopted the K- nearest neighbor (KNN) algorithm, significantly improving the classification performance and achieving a high accuracy rate of 88%.

In addition, there are a series of machine learning-based studies focused on the analysis of other waste datasets. For example, Wu et al. [18] manually extracted partial texture features from garbage images, preprocessed them, and finally classified them using the nearest neighbor method. Gundupalli et al. [19] initially utilized thermal imaging technology to obtain thermal images of electronic waste, then classified garbage images using the Otsu thresholding method. Bonifazi et al. [20] proposed a technique using shortwave infrared hyperspectral images to differentiate between low-density polyethene (LDPE) and high-density polyethene (HDPE) in mixed plastic waste, thereby increasing the utilization rate of construction waste. Xiao et al. [21] classified typical construction waste using near-infrared spectroscopy. Aziz et al. [22] introduced a rotation-invariant solid waste classification, recognition, and detection system. This system determined possible orientations of waste using Huffer curves and classified waste into three categories: "empty," "partially filled," and "filled," using support vector machines. Riba et al. [23] solved the problem of garbage classification by analyzing the infrared spectrum of garbage images and counting multivariate variables. Huang et al. [24] researched the extraction and classification of colour features of construction waste. They identified construction waste using HSV threshold segmentation algorithms and K-means clustering algorithms, with an average processing time of 1.17 seconds. Zheng et al. [14] analyzed the volume of construction waste using the SFS algorithm and classified it using support vector machines.

Although machine learning-based garbage image classification models have achieved some success under specific conditions, their reliance on manually designed features makes it challenging to capture the complex spatial structures and semantic information in images. Furthermore, the limitations of domain knowledge may lead to poor model performance when

dealing with unknown or complex garbage images. In addition, traditional machine learning methods are sensitive to noise, and issues like noise and image quality under suboptimal shooting conditions further degrade the model's classification accuracy.

## Garbage image classification method based on convolutional neural network

With the continuous improvement of computer computing power and the rapid development of the internet, deep learning simulates the information processing mechanism of the human brain, using multi-layer neural networks to automatically extract complex and abstract features from raw data, enabling more accurate data analysis and prediction. Deep learning models typically consist of an input layer (responsible for receiving raw data), hidden layers (accountable for extracting data features), and an output layer (which generates the final prediction or classification result). It has the following advantages:

- End-to-end learning: Deep learning enables end-to-end learning, where raw data is directly inputted to produce the final decision output without requiring intermediate transformation or processing steps [25].

- Automation of feature learning: Traditional machine learning methods typically require manual feature extraction, whereas deep learning can automatically learn valuable features from data, significantly reducing the amount of preprocessing work [25].

- Ability to handle complex tasks: Deep learning models can capture non-linear relationships within data, making them particularly effective for complex tasks such as image and speech recognition and natural language understanding [26].

- Driving force for research and development: Research in deep learning has driven the rapid growth of algorithms, hardware (such as GPUs and TPUs), and optimization techniques, providing momentum for the overall progress of artificial intelligence.

As one of the critical branches of deep learning, convolutional neural networks have made significant progress in various fields, including computer vision, natural language processing, and speech recognition. Since AlexNet [27] won the ImageNet image classification competition, researchers have successfully proposed a series of advanced algorithms by constantly improving the network architecture, increasing the network depth, optimizing the internal connection, and introducing new technologies, including classic architectures such as VGGNet [28], DenseNet [29], GoogleNet [30], MobileNetV2 [31], InceptionV2 [32], and ResNet [33], as well as new models such as iAFPs-Mv-BiTCN [34], AIPs-DeepEnC-GA [35], DeepAVP-TPPred [36], PAtbP-EnC [37], Deepstacked-AVPs [38] and AIPs-SnTCN [39] in recent years. The emergence of these models has incredibly advanced computer vision development and provided a solid theoretical foundation and technical support for applications in other fields.

As a result, convolutional neural networks are widely used in garbage image classification tasks. In 2016, Yang et al. [15] from Stanford University used support vector machines and CNN algorithms to train image classification models on the TrashNet dataset, with classification accuracies of 67% and 82%, respectively. Rabano et al. [40] put forward a new model based on the lightweight neural network MobileNet, which achieved an accuracy of 87.2% on the TrashNet data set. Aral et al. [41] used various networks (including DenseNet121, DenseNet169, MobileNet, Xception and InceptionResNetv2) to classify TrashNet data sets and found that DenseNet121 had the highest accuracy. Kennedy et al. [42] used the transfer learning strategy for reference to introduce the OscarNet model, which achieved a verification

accuracy of 88.42% on TrashNet data sets. Bircanoğlu et al. [43] put forward a novel garbage image classification model(Recycle Net), which shows impressive performance on the Trash-Net data set. Ruiz et al. [44] proposed an automatic garbage image classification algorithm based on ResNet, which achieved an average accuracy of 88.66% on the TrashNet data set. Adedeji et al. [45] use ResNet101 to extract features from the TrashNet dataset and then use a support vector machine instead of a complete connection layer for classification. Ozkaya et al. [46] combined different neural networks with different classifiers, compared their performances, and concluded that the combination of Google Network and SVM classifier produced the best classification result. Shi et al. [47] realized the effective fusion of feature information by widening network branches and increasing layers. Then, they proposed a model called M-b Xception, which was specially used for the classification task of garbage images. Zhang et al. [48] proposed that embedding the self-monitoring module in the residual network model can effectively integrate the relevant features of each channel graph and compress the spatial dimension information, thus significantly improving the representation ability of the feature graph. Ma et al. [49] proposed a new model based on ResNet-50, which was used to classify the TrashNet dataset and proved more accurate and robust than the existing model. Shi et al. [50] proposed a garbage classification model based on the multi-layer hybrid convolutional neural network by adjusting the number of network modules and channel widths, achieving a classification accuracy of 92.6% on the TrashNet dataset. The recyclable waste classification model proposed by Hossen et al. [51] surpassed several state-of-the-art models with an accuracy of 95.01% on the TrashNet dataset and validated the model's reliability through class activation mapping.

In addition, some convolutional neural network-based garbage image classification models were studied based on self-built garbage datasets. Alsubaei et al. [52] developed a deep learning model focused on small-scale garbage waste detection and classification to assist intelligent waste management systems. The model combined an improved IRD model and the AOA arithmetic optimization algorithm for object detection and used a functional linkage neural network for classification. Liu et al. [53] proposed a garbage image recognition model based on transfer learning and model fusion. Experimental results on their self-built dataset showed that the model had better convergence and accuracy. Li et al. [54] addressed the problems of overfitting and poor convergence in traditional image recognition algorithms by proposing a deep learning-based garbage image recognition algorithm. The algorithm overcame overfitting by introducing Dropout, adjusted the parameters of the deep neural network using the Ada-grad adaptive method, and utilized the ReLU activation function to solve the gradient vanishing problem in neural network training.

Although convolutional neural network-based garbage image classification models show great potential, they still have shortcomings. First, garbage image datasets are usually small and imbalanced, which causes the model to overfit during training, thus affecting the model's generalization ability and classification performance. Second, garbage images have diverse and inconsistent visual features, such as varying lighting conditions, occlusions, or object overlaps, which make the feature extraction process more complex and affect the model's performance in different scenarios. Additionally, some models focus too much on classification accuracy, leading to excessive model parameters and significantly reducing detection speed.

## Proposed methods

This paper presents a detailed comparative analysis of the accuracy, loss, and number of parameters of the more popular image classification models on the CIFAR-10 dataset, covering VGGNet, DenseNet, GoogleNet, MobileNet, InceptionNet, and ResNet. The specific results

**Table 1. Comparison of accuracy and loss of different networks on the CIFAR-10 dataset.**

| Network | Epoch | Train | | Val | | Parameters |
|---|---|---|---|---|---|---|
| | | Accuracy | Loss | Accuracy | Loss | |
| AlexNet | 200 | 0.8644 | 0.2673 | 0.7550 | 1.5667 | 9,639,178 |
| VGG-16 | 200 | 0.9777 | 0.1764 | 0.8831 | 0.9599 | 33,638,218 |
| VGG-19 | 200 | 0.9817 | 0.1752 | 0.8892 | 0.8149 | 38,947,914 |
| DenseNet-121 | 200 | 0.9620 | 0.1982 | 0.9074 | 0.5042 | 8,062,504 |
| DenseNet-169 | 200 | 0.9735 | 0.2589 | 0.9146 | 0.4867 | 14,149,480 |
| GoogleNet | 200 | 0.9607 | 0.1960 | 0.9101 | 0.4820 | 6,998,728 |
| MobileNetV2 | 200 | 0.9380 | 0.2190 | 0.8830 | 0.9660 | 3,504,872 |
| MobileNetV3 | 200 | 0.9445 | 0.2012 | 0.8955 | 0.9354 | 5,474,472 |
| InceptionV2 | 200 | 0.9699 | 0.1823 | 0.9114 | 0.4823 | 11,264,111 |
| InceptionV3 | 200 | 0.9754 | 0.2540 | 0.9157 | 0.4763 | 23,851,784 |
| ResNet-34 | 200 | 0.9841 | 0.1755 | 0.9221 | 0.4510 | 21,815,338 |
| ResNet-50 | 200 | 0.9874 | 0.1709 | 0.9226 | 0.4111 | 25,613,514 |
| ResNet-101 | 200 | 0.9852 | 0.1716 | 0.9204 | 0.4205 | 44,596,810 |
| ResNet-152 | 200 | 0.9868 | 0.1650 | 0.9279 | 0.3923 | 60,266,442 |

are shown in Table 1. The analysis results show that the ResNet series of models excel in various indicators, and the number of parameters of the model grows accordingly with the increase of the model depth, resulting in increased hardware requirements and slower computation speed. Based on carefully considering classification accuracy and computational efficiency, this paper selects ResNet-50 as the backbone network of the garbage image classification model, and its specific structure is shown in Fig 1.

In addition, it is well known that the performance of deep learning models is closely related to the size of the training dataset. Due to the small size of the garbage image dataset and the uneven distribution of categories, the effect of directly using ResNet-50 to classify it is not ideal. Therefore, this paper improves the original ResNet-50, and the structure and parameters of the improved model are presented in Table 2.

## Redundancy-weighted feature fusion

The standard ResNet-50 network extracts features from the original image by 7×7 convolution (Fig 2(a)). While this approach can quickly expand the receptive field and extract high-level features, it is weak in capturing local details of the image. In addition, the direct use of a size-able convolutional kernel acting on the original image at shallower layers of the network may

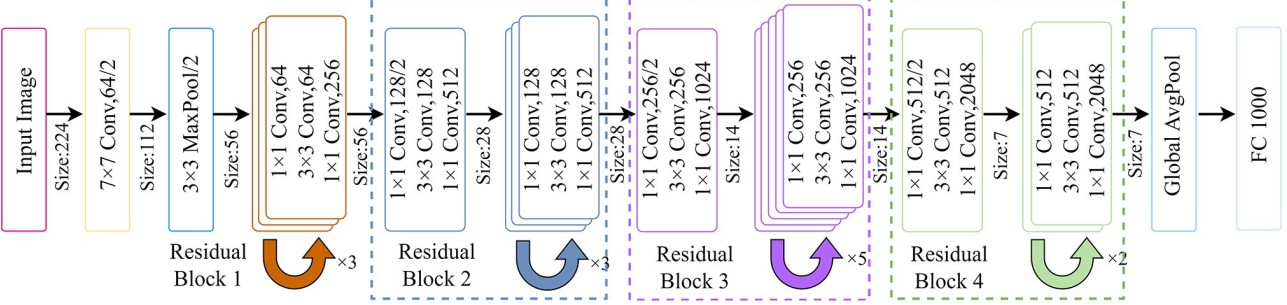

**Fig 1. The detailed structure of the ResNet-50 network.**

**Table 2. The specific structure and parameters of the proposed model.**

| Layer name | Output size | Specific structural parameters |
|---|---|---|
| conv1(redundancy-weighted feature fusion) | 112×112 | $[\alpha(7×7,\text{stride }2);\beta(5×5,\text{ stride }2);\gamma(3×3,\text{ stide }2)]$ |
| conv2_x | 56×56 | 3×3 max pool,stride 2 |
| | | {1×1/64;3×3(depth-separable)/64;1×1/256}×3 |
| conv3_x | 28×28 | {1×1/128;3×3(depth-separable)/128;1×1/512}×4 |
| conv4_x | 14×14 | {1×1/256;3×3(depth-separable)/256;1×1/1024}×6 |
| conv5_x | 7×7 | {1×1/512;3×3(depth-separable)/512;1×1/2048}×3 |
| classification | 1×1 | average pool,6-d fc,softmax |

not be conducive to the gradual construction of complex feature hierarchies by the model. Existing multiscale feature fusion methods (Fig 2(b)) usually adopt two approaches, add and concatenate: add enhances the information content of each dimension without increasing the dimensionality by superimposing the features. In contrast, concatenate increases the dimensionality by feature splicing, but the information content of each dimension remains unchanged. However, the above approaches may introduce a large amount of redundant information, and the features of different scales have their characteristics, so such processing cannot fully feature information of different scales. Therefore, this paper proposes a redundancy-weighted feature fusion module, as shown in Fig 2(c), to improve the initial process of the original ResNet-50 network in the feature extraction stage.

First, this paper employs three parallel convolutional kernels of different sizes (7×7, 5×5, and 3×3) to extract features from the original image. The smaller kernels effectively capture detailed features, while the larger kernels are more suitable for extracting global and macro information. However, these features at different scales often contain a significant amount of

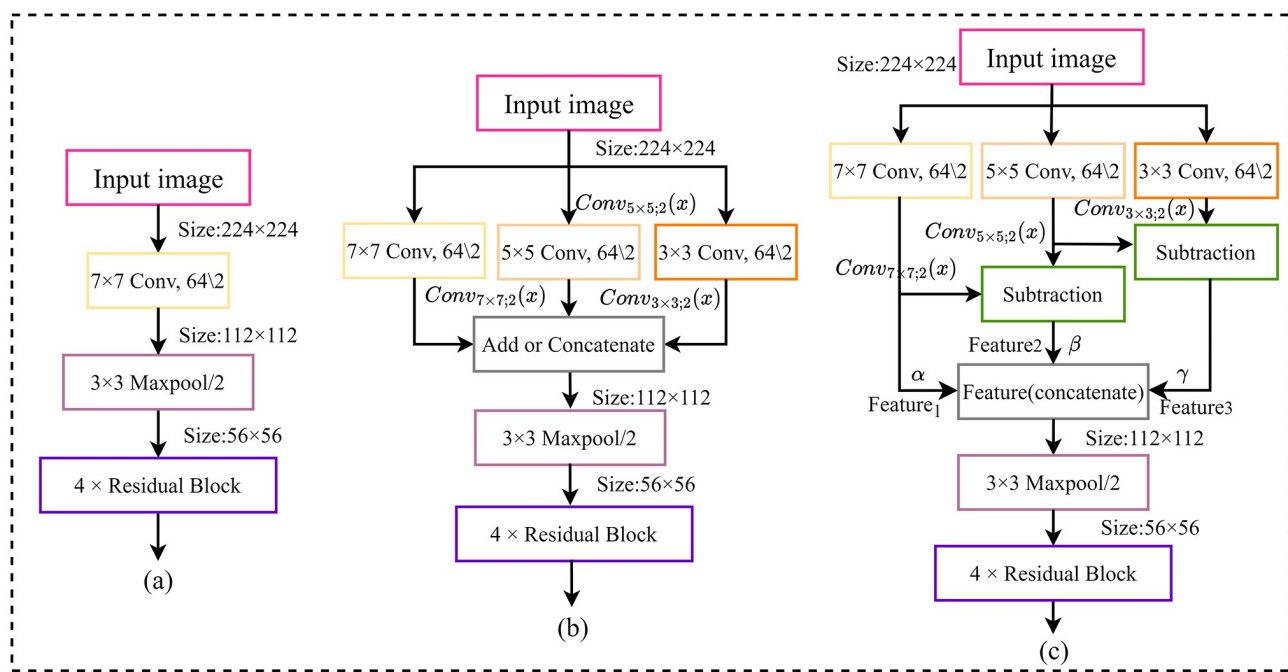

**Fig 2.** (a)shallow feature extraction structure of standard ResNet-50 network. (b)Existing multi-scale feature fusion methods. (c)The specific structure of the redundancy-weighted feature fusion module.

redundant information, and directly using these features may increase the computational cost of the network. Therefore, in this paper, the redundant information is eliminated before the multi-scale feature fusion, and the specific formula is as follows:

$$Feature_1 = Conv_{7\times7;2}(x) \tag{1}$$

$$Feature_2 = Conv_{5\times5;2}(x) - Conv_{7\times7;2}(x) \tag{2}$$

$$Feature_3 = Conv_{3\times3;2}(x) - Conv_{5\times5;2}(x) \tag{3}$$

Where $x$ denotes the original image, $Conv_{7\times7;2}(x)$, $Conv_{5\times5;2}(x)$, and $Conv_{3\times3;2}(x)$ denote the feature maps obtained from the convolution operation on the original image using convolution kernels of different sizes such as 7×7, 5×5 and 3×3, respectively. Since feature maps of different scales contain their unique information, weighting the above feature maps with weight coefficients aims to utilize this information more fully, as in Eq (4).

$$Feature = Concatenate[(\alpha \times Feature_1); (\beta \times Feature_2); (\gamma \times Feature_3)] \tag{4}$$

Here, *Feature* denotes the final multiscale feature fusion tensor, $Feature_1$, $Feature_2$, and $Feature_3$ denote the different scale feature tensors after removing redundant information, respectively, and $\alpha$, $\beta$, and $\gamma$ are their corresponding weight coefficients, collectively known as weight coefficient combinations. The weight coefficients $\alpha$, $\beta$, and $\gamma$ are adaptively learned through backpropagation and gradient descent algorithms in the network training process. Specifically, the network first randomly initializes these weight coefficients. Then, in each iteration, the network multiplies $Feature_1$, $Feature_2$, and $Feature_3$ by their corresponding weight coefficients $\alpha$, $\beta$, and $\gamma$, and adds them together to obtain the *Feature*. Next, the network computes the loss between the predicted results and the truth labels and backpropagates to compute the loss gradients concerning the weight coefficients. Using these gradients, the weight coefficients are updated by applying the learning rate to minimize the loss function. This process is repeated until the network finds an optimal set of weight coefficients that satisfy Eq (5). Through the above design, the redundancy-weighted feature fusion module eliminates the coexisting repetitive information between features of different scales, reduces the computational resource consumption of the network, and, at the same time, ensures that the information related to the current task is thoroughly mined during the feature fusion process, thus enhancing the effectiveness of the model.

$$\alpha + \beta + \gamma = 1 \quad \text{and} \quad \alpha, \beta, \gamma \in [0, 1] \tag{5}$$

## Depth-separable convolution

Deep-separable convolution (Fig 3(b)) consists of channel-by-channel convolution and point-by-point convolution. Unlike the standard 3×3 convolution (Fig 3(a)), channel-by-channel convolution uses a single-channel convolution kernel that acts independently on each channel of the input feature map to generate an output feature map with the same number of channels as the input feature map. However, channel-by-channel convolution may result in too few channels of the output feature map, affecting the adequate representation of information. To solve this problem, point-by-point convolution increases the number of channels by operating on the feature map through a 1×1 convolution kernel, thereby enriching the feature information and improving the representation capability of the network. Assuming that the input feature map is $F' \in \mathbb{R}^{H \times W \times C}$, and the output feature map is $F' \in \mathbb{R}^{H_1 \times W_1 \times C_1}$, the number of

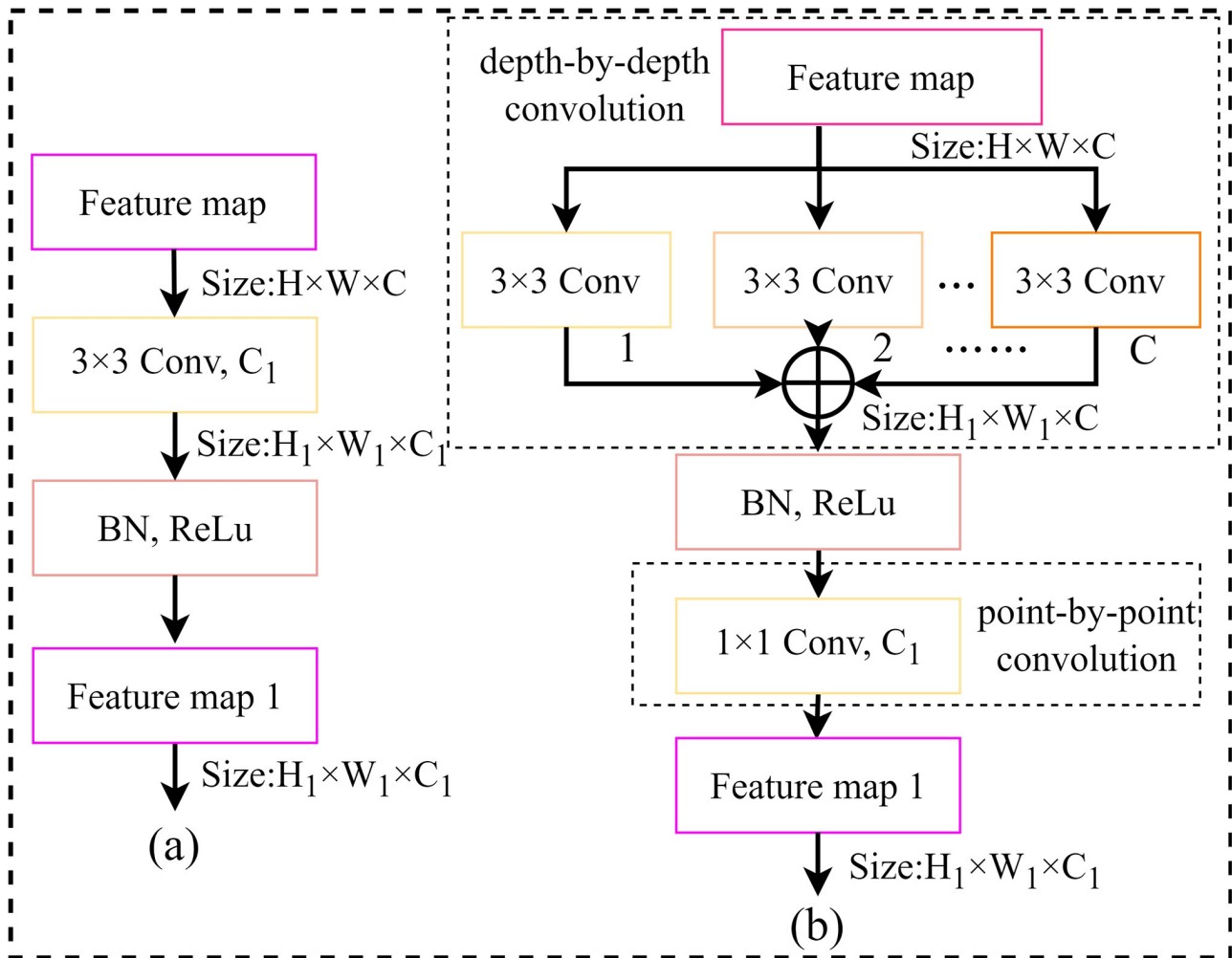

**Fig 3.** (a)Standard 3×3 convolution operation in Resnet-50. (b)Depth-divisible convolution operation.

parameters and the number of computations required for the standard 3×3 convolutional operation in ResNet-50 are given by Eqs (6) and (7).

$$P_s = 3 \times 3 \times C \times C_1 = C \times 9 \times C_1 \tag{6}$$

$$C_s = 3 \times 3 \times C \times C_1 \times H_1 \times W_1 = C_1 \times H_1 \times W_1 \times 9 \times C \tag{7}$$

The number of parameters and the computational effort required for the depth-separable convolution operation are given by Eqs (8) and (9)

$$P_d = 3 \times 3 \times C + C \times C_1 = C \times (9 + C_1) \tag{8}$$

$$C_d = 3 \times 3 \times C \times H_1 \times W_1 + C \times C_1 \times H_1 \times W_1 = C_1 \times H_1 \times W_1 \times (9 + C_1) \tag{9}$$

Compared with the standard 3×3 convolution operation, depth-separable convolution significantly reduces the number of parameters and computation of the model. Therefore, this paper uses depth-separable convolution to replace the standard 3×3 convolution in ResNet-50

to optimize the model structure, reduce the computational burden, and accelerate the inference speed so it performs better in resource-constrained scenarios. In addition, due to the reduction of the number of parameters, the deep separable convolution is less dependent on the data during the training process, which reduces the risk of model overfitting and further improves the model's generalization ability.

## Model loss functions

The vast majority of existing image classification models use the traditional cross-entropy loss function to measure the difference between the actual probability distribution of the samples and the predicted probability distribution and instruct the model to update its parameters during training to reduce this difference and thus improve the accuracy of the prediction. The definition of cross-entropy loss is as follows:

$$L_{\text{cross}} = -\frac{1}{N}\sum_{i=1}^{N}\sum_{c=1}^{C} y_{i,c}\log(p_{i,c}) \tag{10}$$

In Equation Eq (10), $N$ denotes the number of samples, $C$ is the number of categories, and $y_{i,c}$ and $p_{i,c}$ denote the actual probability distribution and probability distribution of sample $i$, respectively. However, the cross-entropy loss assigns the same weight to the loss of each sample, which means that in the case of category imbalance, the model may favour the frequently occurring categories and ignore the characteristics of rare categories, thus weakening the model's ability to recognize the few categories. In order to solve this problem, Focal Loss is developed, which introduces a moderating factor on top of the cross-entropy loss, aiming to enhance the model's focus on the difficult-to-classify samples and thus improve the model's classification performance in the category imbalance scenario. Focal loss is defined as in Equation Eq (11).

$$L_{focal} = -\frac{1}{N}\sum_{i=1}^{N}\sum_{c=1}^{C} y_{i,c}\left(1 - p_{i,c}\right)^{\lambda}\log(p_{i,c}) \tag{11}$$

Where $\lambda$ is a hyperparameter, usually taking the value of 2, used to control how much attention is paid to hard-to-categorize samples. Although Focal loss has been widely used, it is more inclined to focus on the features of hard-to-classify samples and may pay insufficient attention to the features of a few sample categories. In addition, the difficulty of sample classification varies with the dynamic process of model training. For this reason, we weight the Focal loss to enhance the model's ability to learn sparse categories by assigning lower weights to categories with large samples and higher weights to categories with small samples. The model loss function is defined as follows.

$$L = -\frac{1}{N}\sum_{i=1}^{N}\sum_{c=1}^{C} \delta_{i} y_{i,c}(1 - p_{i,c})^{\lambda}\log(p_{i,c}) \tag{12}$$

Where $\delta_i$ is the weighting coefficient, its value is inversely proportional to the number of samples in each class within the imbalanced waste image dataset. In this paper, we apply weighting to Focal Loss to effectively handle difficult-to-classify samples while also alleviating the negative impact of class imbalance on model performance, ensuring the robustness of the entire model.

## Experiments

### Experimental parameter settings and details

In this paper, the Python programming language and PyTorch deep learning framework are used for the construction and training of network models. The experimental equipment includes an AMD Riptide 7 5800X CPU with 64GB of RAM and an NVIDIA RTX 3090 GPU with 24GB of video memory. To fully utilize the GPU performance, the experimental environment integrates CUDA 11.1 and its deep neural network library (CUDNN). During model training, the initial learning rate was set to 1e-3, and the parameters were updated using the AdamW optimizer. To prevent the model from overfitting, the weight decay factor was set to 1e-9. In addition, the learning rate used an exponential decay mechanism that decays to 96% of the original value every 1.3 cycles. The batch size of the whole training process was 12, and the total number of training rounds was 250.

### TrashNet dataset

All experiments in this paper are conducted on the TrashNet dataset, which consists of six categories of RGB images: cardboard, glass, metal, paper, plastic, and trash. The "trash" category includes items that cannot be classified into the other five categories, such as ceramic fragments and rubber products. The dataset contains a total of 2,527 images, each with a resolution of 512×384 pixels. Specifically, the dataset comprises 403 images of cardboard, 501 of glass, 410 of metal, 594 of paper, 482 of plastic, and 137 of general trash. Fig 4 presents representative images for each of the six categories. In this study, each image is resized to 224×224 pixels, and pixel values are normalized to a range of 0 to 1. The dataset is then split into training and validation sets in an 8:2 ratio.

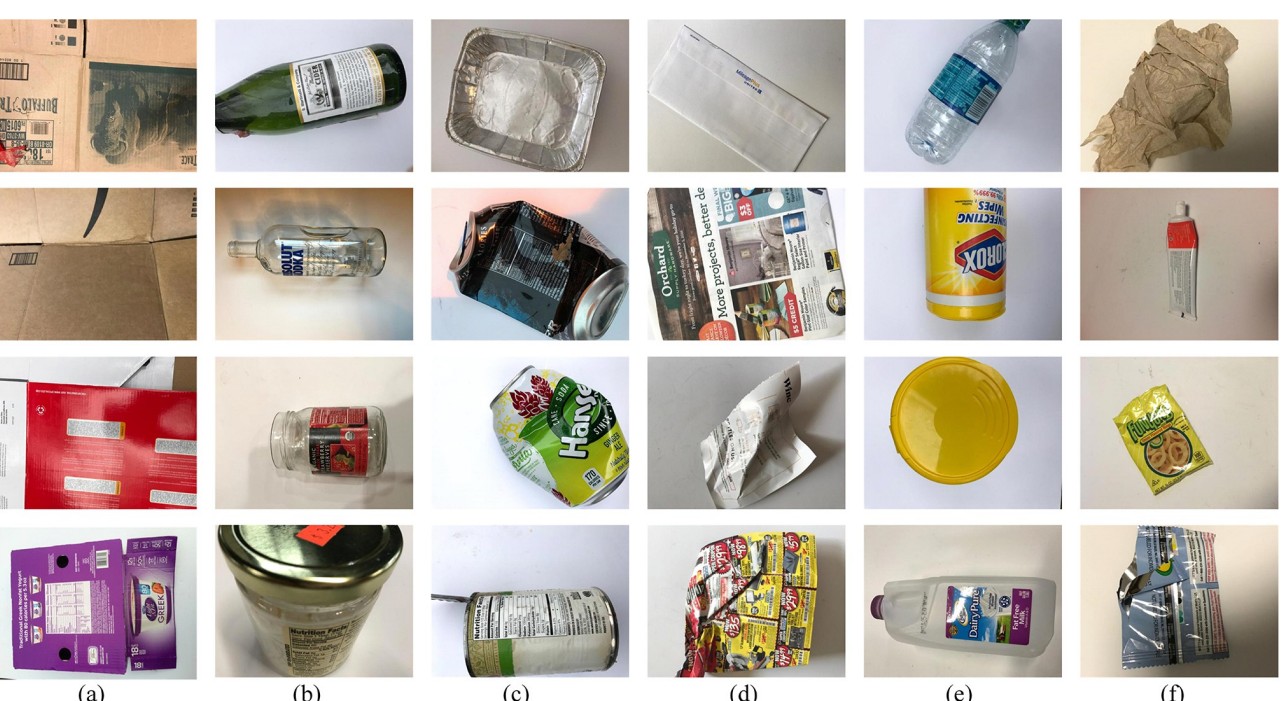

**Fig 4. Different types of typical samples in the TrashNet dataset: (a)cardboard; (b)glass; (c)metal; (d)paper; (e)plastic; (f)trash.**

## Evaluation index

It is well known that the parameters of the model will gradually stabilize as the number of training cycles increases. However, the accuracy and loss will still fluctuate within a small range. Therefore, in this paper, the average accuracy and loss of the last ten training cycles are calculated as the final accuracy and loss. The expressions are as follows.

$$Accuracy\_avg = \frac{\sum_{e=241}^{250} Accuracy_e}{10} \tag{13}$$

$$Loss\_avg = \frac{\sum_{e=241}^{250} Loss_e}{10} \tag{14}$$

Where $Accuracy_e$ and $Loss_e$ denote the classification accuracy and loss of the model on the TrashNet test set after completing the $e-th$ round of training, respectively, and $e$ represents the current training round of the model.

## Experimental results nalysis

First, this study conducted a comprehensive comparative analysis of several popular image classification models with the model proposed in this paper on the TrashNet dataset. The main comparison metrics include classification accuracy ($Accuracy\_avg$), loss ($Loss\_avg$), the time required to validate a single image ($S-IValtime$), and the number of model parameters, as detailed in Table 3. Analyzing the data in Table 3 reveals that, compared to MobileNetV3 [31], which has the fewest parameters, our model significantly improves accuracy on the TrashNet test set by 11.84%. Although the number of model parameters increased by 27,400,940 and the time required to validate a single image also increased by 43.47ms, these increases were necessary to achieve higher classification precision. In comparison to the ResNet-50 [33], which has the highest accuracy, our model achieved a 9.61% increase in accuracy on the TrashNet test set. In this case, the number of model parameters increased by 7,261,898, and the time required to validate a single image only increased by 9.23ms. This indicates that our model maintains good performance in terms of computational resource consumption while achieving high accuracy. In summary, the model proposed in this paper demonstrates significant improvements in classification performance. Although there are increases in the number of parameters and inference time, these costs are reasonable and worthwhile relative to the improvement in accuracy.

In addition, we randomly selected 12 images from the TrashNet test set and used the Grad-CAM technique to generate the corresponding heatmaps, which were then overlaid on the original images to visually demonstrate the areas the model focused on during classification, as

**Table 3. Performance comparison of different models in TrashNet dataset.**

| Method | Input Size | Loss_avg | Accuracy_avg | Parameters | S-I Val time / ms |
|---|---|---|---|---|---|
| AlexNet [27] | 224×224 | 1.8039 | 69.52% | 9,639,178 | 18.56 |
| VGG-19 [28] | 224×224 | 1.3672 | 80.74% | 38,947,914 | 65.76 |
| DenseNet169 [29] | 224×224 | 0.9375 | 82.47% | 14,149,480 | 23.58 |
| GoogleNet [30] | 224×224 | 0.9612 | 82.36% | 6,998,728 | 12.61 |
| MobileNetV3 [31] | 224×224 | 1.1377 | 81.09% | 5,474,472 | 11.32 |
| Inception-v3 [32] | 224×224 | 0.9254 | 82.75% | 23,851,784 | 42.22 |
| ResNet-50 [33] | 224×224 | 0.6539 | 84.52% | 25,613,514 | 45.56 |
| Ours | 224×224 | 0.2871 | 94.13% | 32,875,412 | 54.79 |

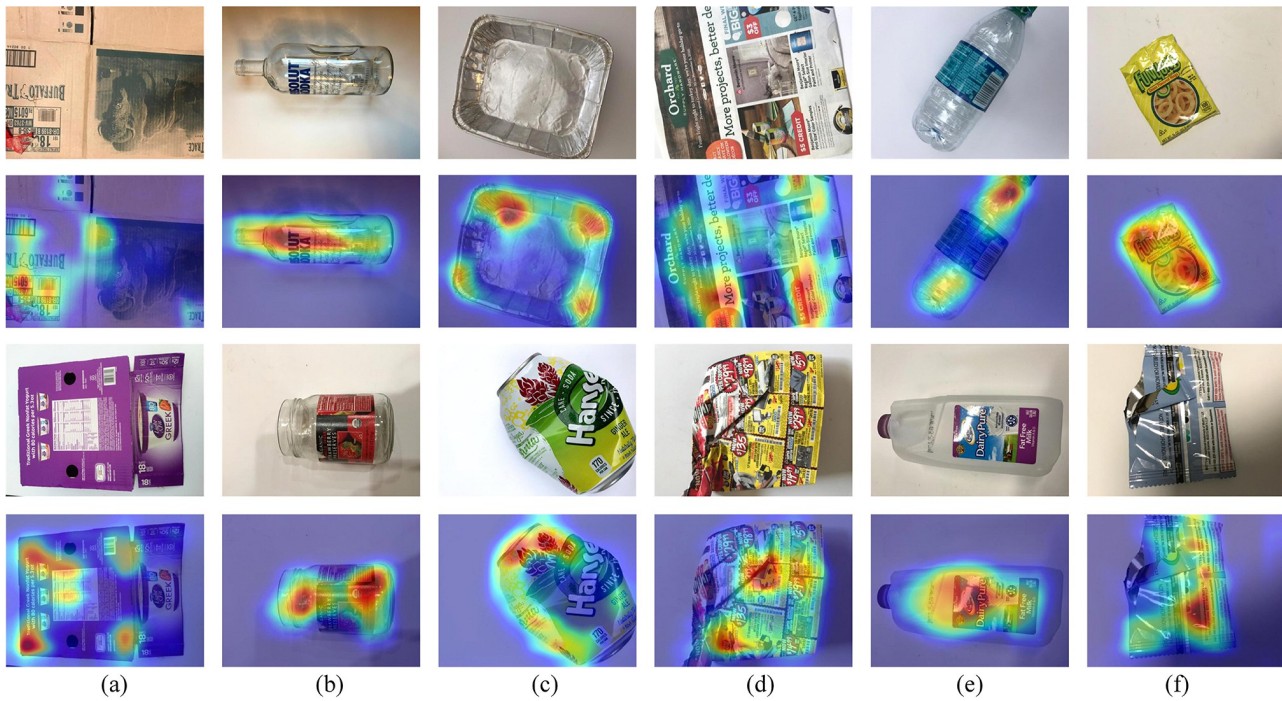

**Fig 5. The heatmap of the model's focused regions based on Grad-CAM.**

shown in Fig 5. In the heatmaps, the red and yellow regions indicate areas that received higher attention from the model. The analysis of the results shows that the proposed model effectively focuses on the objects in the images, demonstrating its ability to capture rich image information and thus achieve high classification accuracy.

This paper also compares the classification accuracies of existing machine learning and deep learning-based garbage image classification models with the model proposed in this paper on the TrashNet dataset, and the results are shown in Table 4. It can be seen that the accuracy of the model in this paper is improved by 6.13% compared to the best-performing machine learning-based model [17]. However, the accuracy is slightly decreased compared to the best-performing model in deep learning [48]. Although there is room for improvement in the accuracy of this paper's model compared to some of the top models, it is worth noting that this paper's model omits the pre-training step, and the overall model size is small. This design not only simplifies the deployment and maintenance of the model but also improves operational efficiency and cost-effectiveness while ensuring higher classification performance. In addition, given that ResNet-50-A [49] and ResNet-50-B [49] also use ResNet-50 as the network backbone, and they are both advanced models proposed in recent years, this paper will further conduct an in-depth comparison and analysis between them and the model in this paper to verify the validity and feasibility of the model in this paper.

To ensure a fair comparison, ResNet-50, ResNet-50-A, ResNet-50-B, and the model proposed in this paper, all utilized the same experimental parameters described in the "Experimental Parameter Settings" section. Fig 6 shows the loss and accuracy curves of the four models over 250 training epochs; it is easy to see that these models gradually stabilize after approximately 120 training epochs, suggesting that the training process is converging and performance is improving. We also compared the models' average accuracy, average loss, the time required for validating a single image, and the number of model parameters with the specific

**Table 4. Comparison of accuracy of different garbage image classification models in TrashNet dataset.**

| Method | Dataset | Backbone | Accuracy |
|---|---|---|---|
| Yang [15] | TrashNet | SVM | 63% |
| Costa [16] | TrashNet | RF | 62.61% |
| Costa [16] | TrashNet | XGBoost | 70% |
| Satvilkar [17] | TrashNet | KNN | 88% |
| Rabano [40] | TrashNet | MobileNet | 87.2% |
| Aral [41] | TrashNet | DenseNet-121 | 89.53% |
| Kennedy [42] | TrashNet | VGG-19 | 88.42% |
| Bircanoğlu [43] | TrashNet | DenseNet-121 | 81% |
| Ruiz [44] | TrashNet | Resnet-50 | 88.66% |
| Adedeji [45] | TrashNet | Resnet-101 | 91.24% |
| Ozkaya [46] | TrashNet | GoogleNt | 90.79% |
| Shi [47] | TrashNet | Xception | 93.68% |
| Zhang [48] | TrashNet | Resnet-18+SVM | 95.37% |
| Ma [49] | TrashNet | Resnet-50 | 88.40% |
| Ma [49] | TrashNet | Resnet-50 | 92.08% |
| Shi [50] | TrashNet | VggNet | 92.6% |
| Hossen [51] | TrashNet | DenseNet201 | 95.01% |
| Ours | TrashNet | Resnet-50 | 94.13% |

results presented in Table 5. Analyzing Table 5 reveals that the accuracy of our model improved by 2.05% compared to the best-performing ResNet-50-B, the number of model parameters decreased by 38,679,954, and the time required to validate a single image was reduced by 9.22 milliseconds. Given that the classification performance of ResNet-50-B is significantly better than that of ResNet-50-A, we further present the confusion matrices for ResNet-50, ResNet-50-B, and our model in the six-class waste classification task in Fig 7. Observing these confusion matrices shows that our model's classification accuracy has effectively improved across all six categories, further validating the superiority and effectiveness of our model.

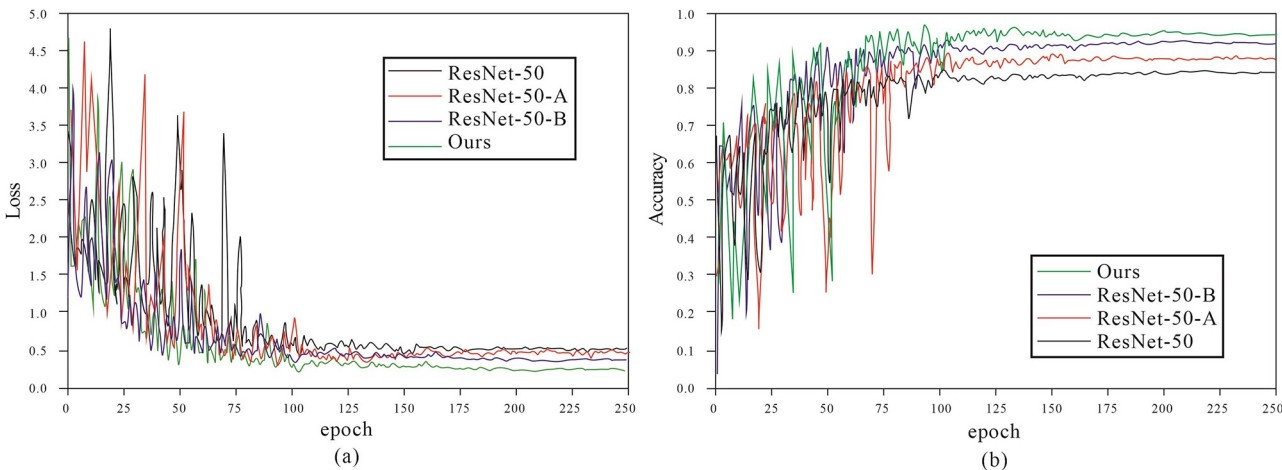

**Fig 6.** (a)Loss curves of different models on TrashNet dataset. (b)Accuracy curves of different models on TrashNet dataset.

**Table 5. The performance of various waste image classification models based on ResNet-50 on the TrashNet dataset.**

| Method | Loss_avg | Accuracy_avg | Parameters | S-I Val time / ms |
|---|---|---|---|---|
| ResNet-50 | 0.6539 | 84.52% | 25,613,514 | 45.56 |
| ResNet-50-A | 0.5243 | 88.40% | 48,965,222 | 60.43 |
| ResNet-50-B | 0.4115 | 92.08% | 71,555,366 | 64.01 |
| Ours | 0.2871 | 94.13% | 32,875,412 | 54.79 |

Additionally, it is well-known that convolutional neural networks are susceptible to external factors. For instance, when the environment around an object changes or the object is partially occluded, the convolutional neural network may make incorrect predictions. In real-world waste classification scenarios, garbage samples often appear in incomplete or obscured forms, making the robustness of the model a vital consideration, as it directly impacts overall classification performance. For this reason, this study divides images into nine regions and applies occlusions to each region, creating nine new test sets (as shown in Fig 8). Subsequently, performance evaluations were conducted on ResNet-50, ResNet-50-B, and our proposed model using these nine new test sets, as illustrated in Fig 9. The results indicate that our model achieved the highest classification accuracy across all nine new test sets. This outcome highlights the robustness and reliability of our model and further validates its adaptability and practical application potential when facing various complex situations.

## Discussion

This paper proposes a new garbage image classification model and comprehensively compares it with the existing machine learning and deep learning-based garbage image classification models on the TrashNet dataset, including several metrics such as accuracy, total model parameters, prediction speed, and robustness. In terms of accuracy, the model in this paper improves by 6.13% compared to the best machine learning-based model. Although the accuracy of this paper's model is slightly less than that of the best-performing deep learning model, it does not require a pre-training process, and the overall model size is small, significantly improving operational efficiency and cost-effectiveness.

In terms of model robustness, to address the susceptibility of convolutional neural networks to external interference, this paper creates nine new test sets by dividing the image into nine regions and masking them one by one to evaluate the model's performance under complex

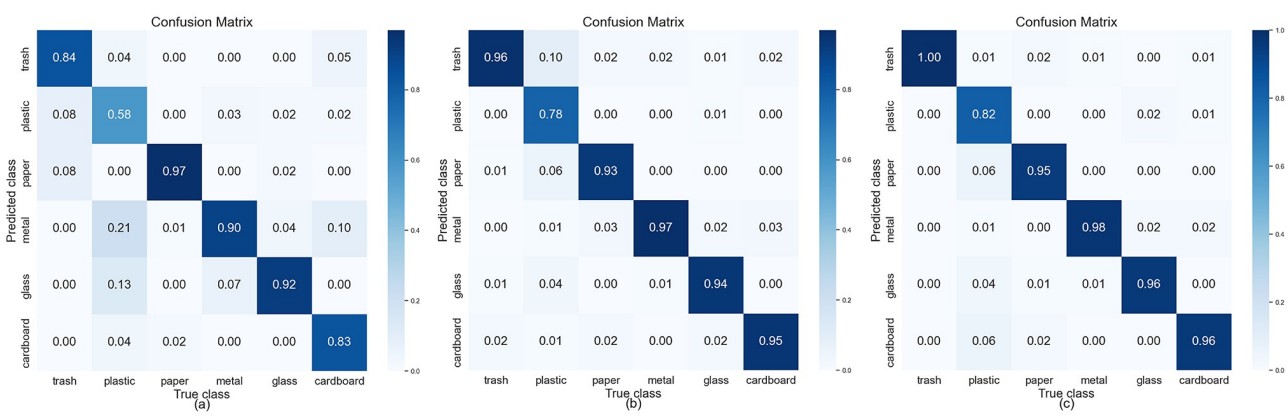

**Fig 7. Confusion matrix of three models in the TrashNet dataset.** (a)ResNet-50 (b)ResNet-50-B (c)Ours.

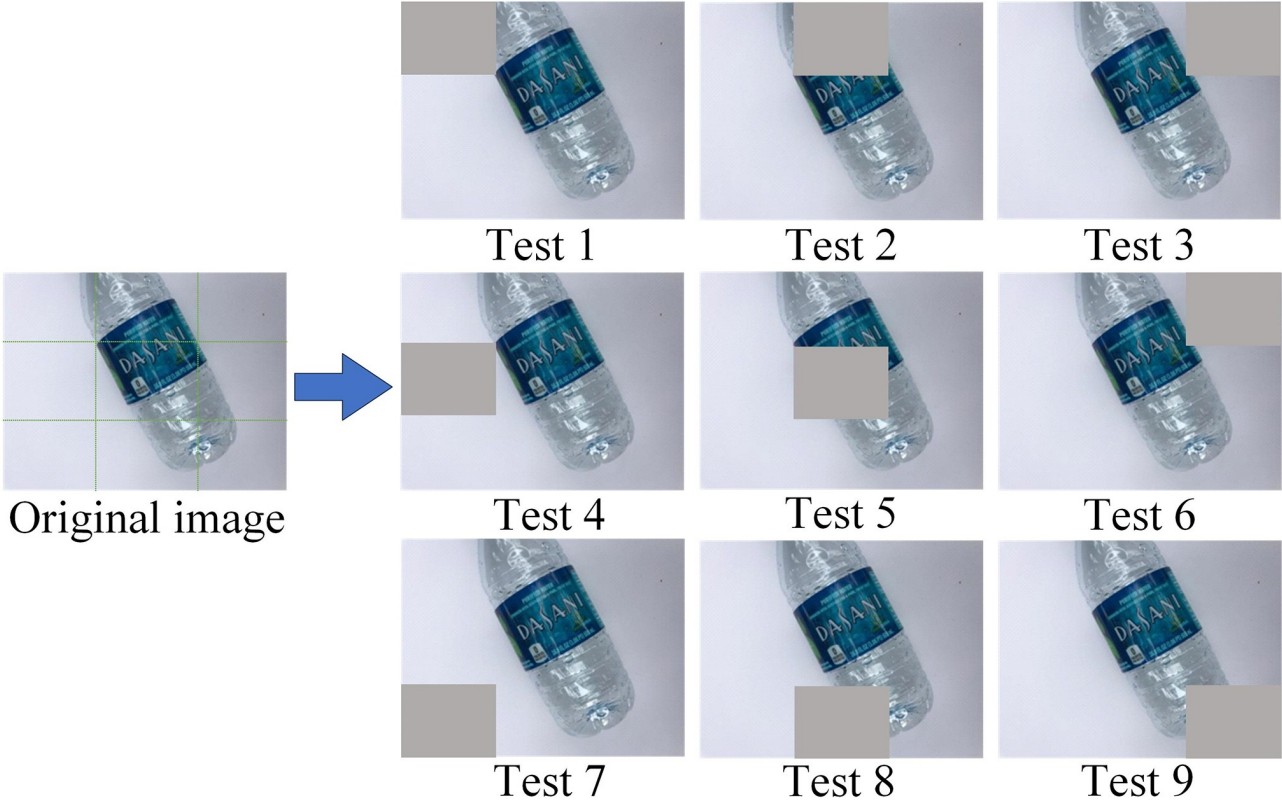

**Fig 8. Randomly masking certain locations of the images in the TrashNet dataset created nine new test sets.**

conditions comprehensively. The results show that the model in this paper achieves the highest classification accuracy on all the new test sets, which further validates its ability to cope with incomplete or occluded samples in real-world garbage classification scenarios and highlights the robustness and reliability of the model.

Although the garbage image classification model proposed in this paper performs well in experiments, it still has some limitations. First, the TrashNet dataset is relatively small and suffers from class imbalance. Although we have mitigated this issue by improving the Focal Loss function, the model may still be influenced by data distribution biases when handling more complex and diverse garbage images in real-world scenarios, leading to overfitting and, consequently, reduced generalization ability. Second, there is a limited number of publicly available garbage image datasets, and this paper only uses the TrashNet dataset for experimentation. As a result, the proposed model may lack sufficient representativeness, and its performance may decline when dealing with different environments or new types of garbage, thus affecting its applicability in real-world situations.

## Conclusion

With the continuous improvement of people's material living standards, the types and quantities of garbage are also increasing rapidly, leading to an increasingly urgent need for garbage classification. Aiming at the problems of low accuracy, insufficient robustness, slow model detection speed, and large scale of existing garbage image classification models, this paper proposes a new garbage image classification model that uses ResNet-50 as the backbone network.

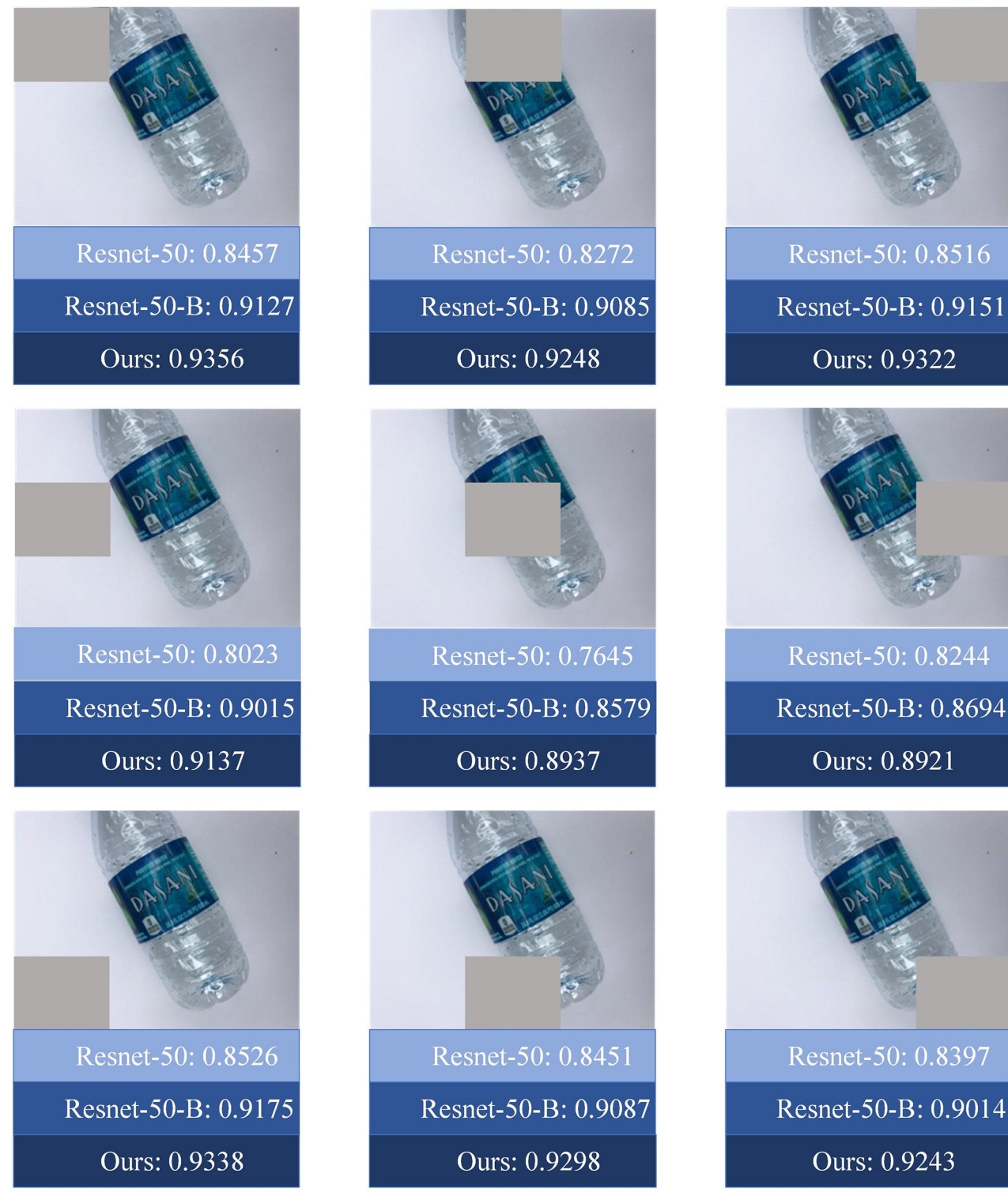

**Fig 9. The accuracy of three models on the nine new test sets.**

Specifically, this paper proposes a redundancy-weighted feature fusion module combined with depthwise separable convolution techniques to optimize ResNet-50. In addition, the standard Focal Loss is weighted to mitigate the impact of class imbalance on model performance. Experimental results on the TrashNet dataset show that the proposed model significantly improves classification accuracy and robustness while maintaining fewer parameters and faster detection speed.

Given the issues of insufficient sample size and class imbalance in garbage image datasets such as TrashNet, future research could explore the use of advanced techniques like Generative Adversarial Networks to synthesize additional training data, particularly for the minority classes in the dataset, in order to further enhance the model's performance. In addition, future work could consider building and expanding a custom garbage image dataset through field collection or collaborative sharing. Such a dataset should cover a broader range of environmental conditions, garbage types, and shape variations, thus providing a solid foundation for training more robust and accurate garbage image classification models.

## Author Contributions

**Conceptualization:** Lingbo Li, Miaojie Zou.

**Data curation:** Fusen Guo.

**Investigation:** Runpu Wang.

**Methodology:** Lingbo Li, Yuheng Ren.

**Software:** Runpu Wang, Miaojie Zou.

**Validation:** Lingbo Li, Runpu Wang, Miaojie Zou, Fusen Guo, Yuheng Ren.

**Writing – original draft:** Lingbo Li, Yuheng Ren.

**Writing – review & editing:** Lingbo Li, Yuheng Ren.

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
