## [Decision Letter · Decision Letter 0]

24 Nov 2024

PONE-D-24-42358Improving the Performance of ResNet-50 Model for Garbage Image Classification: Redundancy-Weighted Feature Fusion and Depth-Separable Convolution StrategiesPLOS ONE

Dear Dr. Ren,

Thank you for submitting your manuscript to PLOS ONE. After careful consideration, we feel that it has merit but does not fully meet PLOS ONE’s publication criteria as it currently stands. Therefore, we invite you to submit a revised version of the manuscript that addresses the points raised during the review process.

Specifically, please make sure the literature review includes most recent relevent works. You may cite the works recommended by the reviewers only if you feel these align well with your work. Moreover, the findings of the study and the contributions may be elaborated further to improve the tutorial content of the manuscript. 

We look forward to receiving your revised manuscript.

Kind regards,

Muhammad Bilal, Ph.D.

Academic Editor

PLOS ONE

Journal Requirements:

2. Please note that PLOS ONE has spec6ific guidelines on code sharing for submissions in which author-generated code underpins the findings in the manuscript. In these cases, all author-generated code must be made available without restrictions upon publication of the work. Please review our guidelines at https://journals.plos.org/plosone/s/materials-and-software-sharing#loc-sharing-code and ensure that your code is shared in a way that follows best practice and facilitates reproducibility and reuse.

Reviewers' comments:

Reviewer's Responses to Questions

**Comments to the Author**

1. Is the manuscript technically sound, and do the data support the conclusions?

Reviewer #1: Partly

Reviewer #2: Yes

Reviewer #3: Yes

2. Has the statistical analysis been performed appropriately and rigorously? 

Reviewer #1: Yes

Reviewer #2: Yes

Reviewer #3: Yes

3. Have the authors made all data underlying the findings in their manuscript fully available?

Reviewer #1: Yes

Reviewer #2: Yes

Reviewer #3: No

4. Is the manuscript presented in an intelligible fashion and written in standard English?

Reviewer #1: Yes

Reviewer #2: Yes

Reviewer #3: Yes

5. Review Comments to the Author

Reviewer #1: Overall the paper is well-organized and interesting. However, the following recommendations are needs to incorporated in order to improve the quality of the paper.

1. In the literature review, the authors are advised to add some updated methods from 2022-2024 to provide an up to date overview.

2.

3. In deep learning section, the authors are advised to add necessary introduction related to deep learning and its importance by citing the recent deep learning models such as iAFPs-Mv-BiTCN, AIPs-DeepEnC-GA, DeepAVP-TPPred, PAtbP-EnC, Deepstacked-AVPs, and AIPs-SnTCN for the readers concern to provide broader overview

4. In order to provide deep understanding of the model, the authors are suggested to provide SHAP analysis based feature analysis to highlight the high contributory features that might be helpful for future real life implementation.

5. How the authors evaluate the overfitting and generalization of the proposed model.

6. An up to date comparison of the proposed model with existing state of the art models will be necessary.

7. What should be the future directions of the proposed model

Reviewer #2: Improving the Performance of ResNet-50 Model for Garbage Image Classification: Redundancy-Weighted Feature Fusion and Depth-Separable Convolution Strategies

#Abstract: The abstract is well composed stating the project problem, a approach to the problem identified, and solution proffered.

# Introduction: There is a need for clarity in the exiting model and the proposed model. In the last paragraph of the introduction section. “The paper proposed ResNet-50 Model.”

In Table 2. “ResNet-50 [33]” indicating the model was used by this author(s). Also, on the same Table there is a row with “Ours”. The question is how do you arrived to “Ours” model? Much should be done in the area of parameters for better understanding to researchers in other field of study.

Also, the issue needs to be redressed for better understanding of the authors contributions to the body of knowledge.

#Figures: There is a need for the authors to improve the figures attached resolutions for clarity.

Reviewer #3: The manuscript proposes an improved ResNet-50 model for garbage image classification, incorporating redundancy-weighted feature fusion and depth-separable convolution strategies. The study aims to address challenges such as low accuracy, robustness issues, and computational inefficiencies in existing models. The authors validate their approach using the TrashNet dataset, demonstrating enhanced performance in terms of accuracy, robustness, and computational efficiency. The work contributes to practical applications in waste management and environmental sustainability.

Comments:

1. The title is clear and relevant but could be more concise by removing redundant phrases like "Improving the Performance of." For example, "Enhanced ResNet-50 for Garbage Classification: Feature Fusion and Depth-Separable Convolutions" might be more impactful.

2. The abstract effectively summarizes the study's objectives and findings but lacks specific quantitative results (e.g., accuracy improvement percentages). Including these would strengthen its impact and clarity.

3. The introduction provides a strong rationale for the study by highlighting the environmental importance of waste classification and the limitations of existing methods. However, it could better articulate the novelty of this approach compared to prior works.

4. The literature review is comprehensive and well-structured, offering a broad overview of related works in machine learning and CNN-based garbage classification. However, it lacks critical analysis of gaps in prior studies that this research addresses.

5. The methodology is detailed and includes innovative contributions such as redundancy-weighted feature fusion and depth-separable convolutions. However, some technical aspects (e.g., how weight coefficients are learned) require further clarification for reproducibility.

6. Results are presented with sufficient detail but could benefit from additional visualizations (e.g., confusion matrices or feature maps) to illustrate model performance across categories. Comparative analysis with state-of-the-art methods should be expanded.

7. The discussion effectively relates findings to research questions but could better acknowledge limitations, such as dataset size or potential overfitting risks due to TrashNet's limited diversity.

8. The conclusion succinctly summarizes key findings but does not provide actionable recommendations or insights into future research directions.

9. References are relevant and formatted correctly but should include more recent studies to reflect the latest advancements in garbage classification.

10. Writing quality is generally good, with minor grammatical errors and occasional verbosity that could be streamlined for better readability.

Major revisions are required before acceptance. Specific areas for improvement include clarifying methodological details, expanding comparative analyses, addressing limitations, refining the title and abstract, and enhancing result visualizations.

6. PLOS authors have the option to publish the peer review history of their article (what does this mean?). If published, this will include your full peer review and any attached files.

Reviewer #1: No

Reviewer #2: No

Reviewer #3: No

---

## [Author Response · Author response to Decision Letter 0]

3 Dec 2024

Dear Editors,

We sincerely thank you for your valuable feedback and the time you spent reviewing our manuscript. We have carefully addressed all the reviewer's comments and revised the manuscript accordingly. We hope that the revised manuscript meets the expectations of the reviewers and the editorial board. Thank you again for your consideration, and we look forward to your response.

Best regards,

Yuheng Ren

---

## [Decision Letter · Decision Letter 1]

9 Jan 2025

Enhanced ResNet-50 for Garbage Classification: Feature Fusion and Depth-Separable Convolutions

PONE-D-24-42358R1

Dear Dr. Ren,

We’re pleased to inform you that your manuscript has been judged scientifically suitable for publication and will be formally accepted for publication once it meets all outstanding technical requirements.

Kind regards,

Muhammad Bilal, Ph.D.

Academic Editor

PLOS ONE

Additional Editor Comments (optional):

The quality of figures depicting the proposed architecture is still very poor. Please improve the presentation of these figures for the final draft.

Reviewers' comments:

Reviewer's Responses to Questions

**Comments to the Author**

1. If the authors have adequately addressed your comments raised in a previous round of review and you feel that this manuscript is now acceptable for publication, you may indicate that here to bypass the “Comments to the Author” section, enter your conflict of interest statement in the “Confidential to Editor” section, and submit your "Accept" recommendation.

Reviewer #1: All comments have been addressed

Reviewer #3: All comments have been addressed

2. Is the manuscript technically sound, and do the data support the conclusions?

Reviewer #1: Yes

Reviewer #3: Yes

3. Has the statistical analysis been performed appropriately and rigorously? 

Reviewer #1: Yes

Reviewer #3: Yes

4. Have the authors made all data underlying the findings in their manuscript fully available?

Reviewer #1: Yes

Reviewer #3: No

5. Is the manuscript presented in an intelligible fashion and written in standard English?

Reviewer #1: Yes

Reviewer #3: Yes

6. Review Comments to the Author

Reviewer #1: The required comments are successfully incorporated and now paper has been significantly improved . No more comments from side

Reviewer #3: The authors punctually responded to the reviewers' comments, adequately motivating any non-responses, and generally improving the quality of the manuscript. The manuscript is well-written, contains interesting information, and is suitable for publication in its present form.

7. PLOS authors have the option to publish the peer review history of their article (what does this mean?). If published, this will include your full peer review and any attached files.

Reviewer #1: No

Reviewer #3: **Yes: **Mehrad Aria

---

## [Editor Report · Acceptance letter]

14 Jan 2025

PONE-D-24-42358R1 

PLOS ONE

Dear Dr. Ren, 

I'm pleased to inform you that your manuscript has been deemed suitable for publication in PLOS ONE. Congratulations! Your manuscript is now being handed over to our production team.

Kind regards, 

on behalf of

Dr. Muhammad Bilal 

Academic Editor

PLOS ONE